# Roles of Glutamate Receptors in Parkinson’s Disease

**DOI:** 10.3390/ijms20184391

**Published:** 2019-09-06

**Authors:** Zhu Zhang, Shiqing Zhang, Pengfei Fu, Zhang Zhang, Kaili Lin, Joshua Ka-Shun Ko, Ken Kin-Lam Yung

**Affiliations:** 1Center for Cancer and Inflammation Research, School of Chinese Medicine, Hong Kong Baptist University, Hong Kong SAR999077, China; 2Department of Biology, Hong Kong Baptist University, Kowloon Tong, Kowloon, Hong Kong SAR 999077, China; 3Golden Meditech Center for NeuroRegeneration Sciences (GMCNS), Hong Kong Baptist University, Kowloon Tong, Kowloon, Hong Kong SAR999077, China

**Keywords:** Parkinson’s disease, glutamate receptors, NMDA receptor, mGluR4, mGluR5

## Abstract

Parkinson’s disease is a progressive neurodegenerative disorder resulting from the degeneration of pigmented dopaminergic neurons in the substantia nigra pars compacta. It induces a series of functional modifications in the circuitry of the basal ganglia nuclei and leads to severe motor disturbances. The amino acid glutamate, as an excitatory neurotransmitter, plays a key role in the disruption of normal basal ganglia function regulated through the interaction with its receptor proteins. It has been proven that glutamate receptors participate in the modulation of neuronal excitability, transmitter release, and long-term synaptic plasticity, in addition to being related to the altered neurotransmission in Parkinson’s disease. Therefore, they are considered new targets for improving the therapeutic strategies used to treat Parkinson’s disease. In this review, we discuss the biological characteristics of these receptors and demonstrate the receptor-mediated neuroprotection in Parkinson’s disease. Pharmacological manipulation of these receptors during anti-Parkinsonian processes in both experimental studies and clinical trials are also summarized.

## 1. Introduction

Parkinson’s disease (PD) is a debilitating neurodegenerative disorder which is second to Alzheimer’s disease as the most common age-related disease. The clinical symptoms of PD include motor disturbances (such as resting tremor, bradykinesia), rapid eye movement behavior disorder, as well as autonomic and cognitive impairment [1]. The pathology underlying PD comprises the degeneration of dopaminergic neurons in the substantia nigra pars compacta (SNc) and the accumulation of intracytoplasmic inclusions, which are known as Lewy bodies in these neurons [2]. Therefore, dopamimetic drugs, including the dopamine precursor levodopa (l-3,4-dihydroxyphenylalanine, l-DOPA), and dopamine receptor agonists are currently considered as the only standard therapy for treating Parkinsonian symptoms [3]. Although these treatments ameliorate the motor signs of PD for several years in most patients, prolonged therapy frequently leads to the development of motor complications, known as L-DOPA-induced-dyskinesia (LID), such as choreic or larger amplitude choreo-athetotic movements, dystonia, and ballism [4]. It is known that different neurotransmitter systems in the human brain and central nervous system (CNS) are involved in the pathophysiology of PD and LID. Among these, glutamate takes up 40% of all synapses and plays an important role in the mediation of basal ganglia circuitry in continuous feedback, leading to the dopaminergic denervation of the striatum [5]. Moreover, an increasing body of evidence has demonstrated the contribution of glutamatergic transmission to the processes of PD and LID [4]. It was also determined that the concentration of serum glutamate in PD patients is higher than that in healthy subjects [6]. Therefore, it has been suggested that pharmacological therapies with the potential to restore normal glutamatergic functions show promise as therapeutic interventions by reversing the severe motor complications that derive from the current dopamine replacement strategies.

Glutamate plays a critical role in brain function through multiple receptor proteins which are primarily located on pre- and post-synaptic neurons in virtually all areas of the CNS. Figure 1 shows that glutamate receptors were originally classified into two major classes of ionotropic (iGluRs) and metabotropic receptors (mGluRs), according to pharmacological means. The iGluRs are multimeric ion channels and are responsible for fast excitatory transmission in the mammalian CNS. Through binding the presynaptically released glutamate, iGluRs transduce signals into the excitation of postsynaptic neurons on a millisecond timescale. This process generates a synaptic current crucial to brain function and regulates learning and memory. iGluRs can be further classified into N-methyl-d-aspartate (NMDA) receptors, α-amino-3-hydroxy-5-methyl-4-isoxazolepropionic acid (AMPA) receptors, and kainate receptors. As members of the G-protein receptors superfamily, the mGluRs mediate slow glutamate responses, which contributes to long-lasting changes in synaptic activity [7]. The mGluRs family can be further divided into eight receptor subtypes. Based on their sequence homology, signal transduction mechanisms, and pharmacological profile, these subtypes are classified into three groups. Group I receptors include mGluR1 and mGluR5, which are linked to phospholipase C-mediated polyphosphoinositide hydrolysis, while group II includes mGluR2 and mGluR3, which in recombinant systems are negatively coupled to adenylate cyclase. Group III consists of mGluR4, −6, −7, and −8, which are also negatively coupled to adenylate cyclase or linked to ion channels [8]. In the pathophysiology of PD, there are regulatory alterations of glutamate receptors in specific loci in the basal ganglia. In addition, glutamate receptors are also changed in the process of LID, such as the increased specific binding of NMDA receptors and decreased mGlu2/3 receptors after levodopa treatment in 1-methyl-4-phenyl-1,2,3,6-tetrahydropyridine (MPTP)-lesioned monkeys [9]. 

Therefore, a comprehensive understanding of glutamate receptors, concerning both the pathophysiology and the treatment targets of PD, may contribute to the development of novel therapeutic approaches to PD. In this review, we discuss the distribution of these different subtypes of glutamate receptors and their neuroprotective properties, as well as evidence of the pharmacological manipulation of these receptors in PD. 

## 2. Basic Biology and Alterations of Glutamate Receptors in PD

Most studies of these receptors are initially focused on their distribution, structure, and complex subunits, as well as functions. However, these factors would change due to dopamine depletion in PD.

### 2.1. Basic Biology and Alterations of iGluRs in PD

NMDA receptors have been the subject of intense study and are the best characterized of the iGluRs. They are widely expressed in the basal ganglia and directly mediate the excitation of neurons in the striatum, globus pallidus, subthalamic nucleus, substantia nigra reticulata, and substantia nigra compacta [10]. These receptors consist of seven subunits: GluN1, GluN2A, −B, −C, and −D, as well as GluN3A, −B, and −C, which assemble in the form of heterodimers or heterotrimers. Because of their high permeability to calcium ions and their ability to trigger a cascade of downstream calcium-dependent signal-transduction processes of physiological and pathophysiological changes, NMDA receptors play an important role in the regulation of excitatory synaptic transmission. Ligands need to occupy both the glutamate site and the cofactor glycine site for the activation of an NMDA receptor [11]. A glutamate site is a binding pocket on the GluN2A and GluN2B subunits formed through interactions between the N-terminus of the receptor and the extracellular loops. Cofactor glycine binding sites exist in the homologous region of the GluN1 and GluN3 subunits and form an excitatory glycine receptor upon heteromeric assembly. At normal resting membrane potential, the ion channel is blocked by the ambient extracellular magnesium through their binding to a site within the pore, which makes the sites of glutamate and glycine appear occupied [12]. On the other hand, due to the voltage-dependent property of the channel induced by the magnesium blockade, excessive stimulation of NMDA receptors by agonists, such as glutamate and glycine, could induce the depolarization of neurons and lead to the extrusion of Mg^2+^ from the channel. This allows the ions to flow inward, especially for a Ca^2+^ influx, by opening the ion channel pore. The accumulation of Ca^2+^ in cytoplasmic material triggers a series of Ca^2+^-dependent enzymes and pathophysiological changes in intracellular biochemistry, which may lead to neuronal damage and death. This biofunction on the regulation of both cellular excitability and biochemistry indicates that NMDA receptors play a major role in excitatory synaptic transmission, plasticity, and excitotoxicity [13,14]. It was reported that both PD-induced dopamine depletion and L-DOPA treatment lead to the redistribution of NMDA receptor subunits. For example, the GluN2A subunits and GluN2A/GluN2B subunit ratio were increased in PD patients as well as in levodopa-treated dyskinetic rats and monkeys [15]. Despite the different sensitivities of subunits to dopamine depletion, the expression levels of GluN1 and GluN2B were augmented in the distinct surface compartment of PD rats [16]. A recent study also showed the increased GluN2D subunit in the striatum of L-DOPA-treated Parkinsonian rats [17]. A growing body of literature has shown substantially increased NMDA-sensitive glutamate binding in the striatum and nucleus accumbent in both experimental models of PD and tissue of PD patients, which may accelerate the degenerative process [18]. Therefore, considerable evidence shows that NMDA receptors are viable targets for the treatment of PD and that their antagonists can provide anti-Parkinsonian benefits in various animal models of PD.

AMPA receptors are postsynaptic ion channels that regulate the majority of the fast excitatory amino acid neurotransmitters in the brain. They are cation-selective heterooligomers assembled in various combinations of the subunits GluR1, −2, −3, and −4 (or GluR A, −B, −C, and −D). AMPA receptor subunits are expressed abundantly and localized in different sites within neuronal cell bodies and processed in the cerebral cortex, basal ganglia, limbic system, thalamus, cerebellum, and brainstem, with precise patterns in the different brain regions [19]. In contrast to NMDA receptors, AMPA receptors are weakly permeable to external Ca^2+^. When gated open by synaptically released glutamate, AMPA receptors mediate the inflow of Na^+^ (and in some cases Ca^2+^) into neurons, while they cause the efflux of K^+^. Although the ion channels assembled by the homomeric or heteromeric combinations of GluR1 and GluR3 show an appreciable Ca^2+^-permeability, the incorporation of a GIuR2 subunit suppresses Ca ^2+^ permeability [20]. It has also been demonstrated that the activation of AMPA receptors increases the influx of extracellular Zn^2+^ into nigral dopaminergic neurons, which leads to movement disorder [21]. This suggests that Ca ^2+^ and Zn^2+^-permeable GluR2-lacking AMPA receptors are particularly important for the treatment of PD. In the 6-OHDA rat model, AMPA receptor-mediated excitatory transmission was enhanced, which increased the release of glutamate [22]. Also, the elevated expressions of AMPA receptors in Parkinsonian monkeys induced using MPTP and dopomimetic treatment have been observed in subregions of the striatum [23]. In addition, a previous study noted the upregulation of AMPA glutamate receptors in the lateral putamen of advanced PD patients experiencing levodopa-induced motor complications (both abnormal involuntary movements and “wearing-off” phenomenon) when compared to others without motor complications [24]. Therefore, antagonists of AMPA receptors could show potential anti-Parkinsonian effects.

The progress in the study of the characteristics and functions of kainate (KA) receptors is lagging more than that of NMDA and AMPA receptors due to the lack of selective kainite receptor (KAR) antagonists and agonists. Kainate receptors are also ligand-gated channels with permeability to cations. They consist of subunits of Gluk1, −2, −3 (previously called GluR5, −6, and −7), and −4, −5 (previously known as KA1 and KA2). KA receptors are highly expressed in the CNS, with the distribution of all subunits occurring in the neocortex [25]. Cells from different regions express different subunits of KA receptors, which is consistent with the fact that most GluK2 and GluK5 subunits are distributed in the rodent cortex [26]. According to the measurement of affinity for kainate receptors, the affinity of GluK1, GluK2, and GluK3 subunits is low, while that of subunits GluK4 and GluK5 is high [27]. Among all the subtypes, GluK4 and GluK5 assemble homomeric or heteromeric receptors with subunits of GluK1 to GluK3 to form functional channels. This is because subunits with a high affinity cannot form functional channels, so they co-assemble with low-affinity subtypes. Although kainate receptors do not cross-assemble with AMPA receptors, there are still many common characteristics between them. However, kainate receptors, found in both intracellular and extrasynaptic distribution in the striatum, share more common features with metabotropic receptors than with ionotropic receptors, which suggests that kainate receptors could exert functions of metabotropic receptors in the striatum [28]. KARs exist in fewer synapses and physiological studies have suggested that KARs mainly contribute to the modulation of synaptic transmission, which is different from the functions of NMDA and AMPA receptors. The latter are the main postsynaptic targets for glutamate that is released synaptically. For example, it was reported that presynaptic KAR activation inhibits GABAergic outflow from the striatum in the rat globus pallidus [29], which may be helpful for improving the treatment of PD, since increased GABAergic activity from the striatum to the globus pallidus is known as one of the cardinal features of PD pathophysiology. KARs also contribute to both short- and long-term synaptic plasticity in the hippocampus, as well as the sensory cortex [30,31]. It is worth noting that loss of parkin function leads to the increased expression of KAR, which may have a pathogenetic role in parkin-related autosomal recessive juvenile parkinsonism [32].

### 2.2. Basic Biology and Alterations of mGluRs in PD

The group I mGluRs consists of mGluR1 and mGluR5, with the main distribution in the postsynaptic position of dendrites and spines. In the primate striatum, they are also found in the presynaptic sites of nigrostriatal dopamine terminals [33]. The expressions of mGluR1 in cerebellar Purkinje cells, striatonigral, and striatopallidal projection neurons and striatal interneurons are most intensive [34]. A longitudinal positron emission tomography study showed the dynamic changes in the expression of mGluR1 that accompany pathological progression in a PD model of rats which mimics clinical pathology [35]. Intense expression of mGluR5 has been found predominantly in telencephalic regions, such as the cerebral cortex, hippocampus, subiculum, main and accessory olfactory bulbs, anterior olfactory nucleus, olfactory tubercle, striatum, nucleus accumbens, and lateral septal nucleus [36,37]. Recently, a positron emission tomography study showed that the expression of mGluR5 was increased in strategic dopaminergic brain regions of PD patients [38]. Moreover, the genetic knockdown of mGluR5 was determined to decrease LID in an aphakia mouse model of PD [39]. In addition, the binding potential of mGluR5 receptor was also decreased in the 6-OHDA rat model of PD [40]. In physiological conditions, the acute regulation of mGlu5 receptor signaling in cortical astrocytes causes oscillatory changes of Ca^2+^ and the synaptic release of neurotransmitters, as well as the programming of transcriptional events [41]. This effect on Ca^2+^ may bring about the interaction between mGlu5 receptors and NMDA receptors. It was reported that mGlu5 receptors enhanced NR2B-containing NMDA receptor activity in the rat hippocampus through the phosphorylation of NR2B [42]. Therefore, a negative allosteric modulator of group I mGluRs, especially for mGlu5 receptors, would be promising to show anti-Parkinsonian effects through the reduction of the excitatory drive in the overstimulation basal ganglia nuclei. 

Group II mGluRs, including mGluR2 and mGluR3, are differentially distributed. The expression of mGluR2 is less than that of mGluR3in the CNS. mGluR2 is mainly expressed in Golgi cells in the cerebellar cortex, mitral cells of the accessory olfactory bulb, the external part of the anterior olfactory nucleus, and some neurons in the entorhinal and parasubicular cortices [43]. mGluR3 is widely distributed in the olfactory tubercle, dentate gyrus, cerebral cortex, nucleus accumbens, lateral septal nucleus, striatum, amygdaloid nuclei, cerebellar cortex, and substantia nigra pars reticulata [44]. Functionally, the activation of group II mGluRs regulates corticostriatal synapses and inhibits the generation of thalamus-derived glutamate as well as dopamine from the ventral midbrain, through a direct or indirect presynaptic manner [45]. A recent study showed that expressions of mGluR2/3 protein decreased in 6-OHDA-lesioned rats [46]. A mechanism study also revealed that the activation of mGluR2/3 restored the clearance of extracellular glutamate that was destroyed in astrocytes [47]. Therefore, it has been suggested that selective agonists or positive allosteric modulators of group II mGluR may contribute to PD treatment, due to their contribution to the attenuation of glutamate levels [48]. 

Group III receptors consist of mGluR4, −6, −7, and −8, which are distributed throughout the basal ganglia circuitry. Among group III mGluRs, the expression of mGluR7 is the highest, with extensive distribution in the hippocampus, thalamus, neocortex, amygdala, hypothalamus, and locus coeruleus, whereas that of mGluR6 is predominantly limited to the retina [49,50]. The location pattern of mGluR8 in the CNS has been found at the presynaptic level in the cerebellum, olfactory bulb, hippocampus, and cortical areas, which is also more restricted than that of mGluR7 [51]. Although mGluR4 is mainly distributed in the cerebellum, it also has been proved to be located in other areas, such as the cerebral cortex, olfactory bulb, hippocampus, lateral septum, septofimbrial nucleus, striatum, thalamic nuclei, lateral mammillary nucleus, pontine nuclei, and dorsal horn [52]. In line with the intense expression, mice lacking mGluR4 show a clear impairment in their ability to learn complex motor tasks [53]. Moreover, the activation of mGluR4 may have a neuroprotective effect through the inhibition of glutamate production in the substantia nigra and the reduction of inflammatory effects [54]. Positive allosteric modulators of mGluR4 have been proposed for the symptomatic management of PD and they have been shown to induce a reduction of motor symptoms in animal models of PD. For example, studies have demonstrated that positive allosteric modulators of mGluR4 provide functional neuroprotection against nigrostriatal impairment induced by 6-OHDA in rats and MPTP in macaques [55,56]. Although studies of the anti-Parkinsonian and neuroprotective properties of mGluR6 and mGluR7 are fewer than those addressing mGluR4, these receptor subunits have been proven to increase the proliferation of neural stem cells (NSCs) by promoting cell cycle progression and inhibiting the apoptosis of NSCs [57,58]. NSCs are able to self-renew as well as practice differentiation into the neurons and glia of the nervous system. Many studies have transplanted NSCs to promote the restoration of function after spinal cord injury [59]. Therefore, targeting mGluR6 and mGluR7 on NSCs may also be benefit for the stem cell-based strategy for the treatment of PD. 

## 3. Key Targets of Glutamate Receptors in PD Treatment

The treatment of PD through targeting glutamate receptors can be summarized in three observations: (1) the improvement of PD motor symptoms, (2) the increase of the anti-Parkinsonian efficiency of dopaminergic agents, and (3) the protection of nigral neurons. 

### 3.1. Key Targets of iGluRs in PD Treatment

The antagonists of NMDA reporters include competitive antagonists (e.g., SDZ 220–581, MDL 100, 453), noncompetitive antagonists (e.g., MK-801, dextrorphan, PD 174494, CP-101, 606) as well as glycine site antagonists (e.g., MRZ 2/570, L-701,324, 7-chlorokynurenate, (R)-HA-966) [60]. Plenty of studies have demonstrated that NMDA receptor antagonists can attenuate catalepsy and counteract Parkinsonian rigidity induced by dopamine receptor antagonists in rats as well as akinesia and other motor complications in monoamine-depleted rodents [61,62,63,64]. Similar effects were also shown in an ifenprodil-treated MPTP-lesioned marmoset model [65]. In addition, subthreshold doses of several NMDA receptor antagonists have been determined to synergistically increase the anti-Parkinsonian efficiency of levodopa and other drugs, such as 7-nitroindazole and opioid glycopeptide lactomorphin, as well as prophylactically prevent the appearance of motor response alterations associated with chronic levodopa administration LID both in rat and primate models of PD [66,67,68]. This suggests that these drugs may act as adjunct therapies for improving the efficacy and tolerability of dopaminergic therapies, as well as the combination between different antagonists. On the other hand, the extensive expression of NMDA receptors and widespread excitation stimulated by NMDA has resulted in concern that the global inhibition of these receptors could lead to various unwanted side effects, such as ataxia, impaired learning, and psychosis [69]. Therefore, more and more research is focused on the screening of a series of NMDA subunit-selective drugs. NR2B was demonstrated to be heavily distributed in the striatum and other basal ganglia regions [70,71]. This suggests that therapy targeting NR2B-containing receptors can contribute to more specific effects influencing the NMDA receptor regulation in brain regions related to PD pathophysiology. This was determined by Ifenprodil and traxoprodil (CP-101, 606), antagonists of the NR2B subunit-containing NMDA receptor, which were shown to ameliorate Parkinsonian symptoms and reduce LID in rodents and MPTP-lesioned monkeys [72,73]. The combination therapy of NR2B-selective antagonists with other drugs was also applied. For example, the combination of the GluN2B-selective antagonist Radiprodil and the A_2A_ antagonist Tozadenant led to significant motor improvement both in 6-OHDA-lesioned rats and MPTP-lesioned marmosets [74,75]. In addition, a series of photo-switchable antagonists for the modulation of NMDA receptors have been synthesized by researchers. Among these, PNRA showed selectivity for GluN2A and GluN2C rather than GluN2B, which should be addressed in further studies in more advanced biological settings for potential application in PD treatment [76].

The anti-Parkinsonian properties of AMPA antagonists have been evaluated in different animal PD models. 2,3-Dihydroxy-6-nitro-7-sulfamoyl-benzo[f]quinoxaline-2,3-dione (NBQX), a selective antagonist of AMPA receptors, has been reported to suppress muscular rigidity in monoamine-depleted rats and produce clinically apparent improvement motor deficits in MPTP-lesioned aged Rhesus monkeys [77]. However, many studies have reported that AMPA receptor antagonists are not effective on anti-Parkinsonian action in animal models when given alone. For example, NBQX fails to reverse motor deficits in 6-OHDA-lesioned rats when administered alone [78,79]. Another interesting finding in the same research was that, during co-administration, AMPA receptor antagonists such as NBQX synergistically ameliorated Parkinsonian symptomatology with levodopa in 6-OHDA-lesioned rats and MPTP-lesioned common marmoset. This suggests that AMPA receptor antagonists may be used as adjuvants of levodopa for the improvement of treatment efficacy. In addition, the development of LID promotes the expression and activity of AMPA receptors, which suggests that the blockage of AMPA receptors may alleviate LID therapeutically [24,80]. For example, after administration of the noncompetitive AMPA receptor antagonist LY300164 (talampanel) for levodopa-treated Parkinsonian monkeys lesioned with MPTP, the motor activities were potentiated and LIDs were decreased [23]. In addition, although perampanel, a novel AMPA receptor antagonist, was determined to penetrate the blood–brain barrier, adjunctive Perampanel had no effect on improving LID in PD patients [81,82]. 

At present, several evaluations of pharmacological properties have identified limited compounds that are promising as selective antagonists of kainate receptors, when most target GluK5. For example, compounds of the quinoxalinedione family, such as CNQX and NBQX, play the role of competitive antagonists with affinity for both native and recombinant kainate receptors [26]. Usually, these antagonists are shared with AMPA receptors and are far more effective on AMPA receptors than that on KA receptors. Therefore, some pyrrolyl-quinoxalinedione derivatives with a more potent affinity for kainate receptors have been developed. Take LU97175 for example—it was more selective towards kainate receptors, especially for the sites of kainite subunits with low affinity (i.e., GluK1 to GluK3), which showed anticonvulsant effects without the induction of motor dysfunction [83]. In addition, glutamate receptor antagonists from a new series of 6-substituted decahydroisoquinoline, such as LY382884 and LY377770, display higher affinities with kainate receptors containing the GluK1 subunit than with AMPA and NMDA receptors [84,85]. In addition, due to the global distribution of AMPA/NMDA receptors, the use of AMPA/NMDA receptor antagonists in humans may lead to severe side effects, which is worse because of their fast excitatory effects. In contrast, KARs mainly affect slower excitatory modulation, which is more like the function of mGluRs [29]. This may be achieved through the mediated role of KARs in overactive striatopallidal GABAergic synapses and glutamatergic subthalamopallidal transmission [29], suggesting that KARs represent a huge potential target for the development of a novel therapeutic strategy for PD. However, almost all of the abovementioned studies on GluK1-selective antagonists with promising pharmacological efficiency focus on animal models of pain, migraine, epilepsy, stroke, and anxiety, rather than that of PD. Therefore, direct evidence of the effect of KA receptor antagonists on anti-Parkinsonian action is urgently needed. In addition, novel selective antagonists of other subunits of KA receptors need further development.

### 3.2. Key Targets of mGluRs in PD Treatment

mGluR5 receptors have been reported to be efficiently suppressed by a series of compounds such as phenylpyridine derivatives (2-methyl-6-(phenylethynyl)-pyridine (MPEP), 3-[(2-methyl-1,3-thiazol-4-yl)ethynyl]pyridine (MTEP), AFQ056/mavoglurant, fenobam, and ADX48621/dipraglurant), which are systemically active [86]. Consistent with this hypothesis, several studies have proven that these antagonists of the mGluR5 receptor exert anti-Parkinsonian effects in different animal models of PD. For example, both MPEP-treated and mGluR5 knockout mice displayed a higher survival probability and a reduced extent of nigrostriatal damage induced by the administration of the dopaminergic neurotoxin MPTP, which was also mimicked by SIB1893 [87]. Furthermore, MPEP also alleviated LID in 6-OHDA-lesioned rats [88]. With higher solubility than MPEP, MTEP was also proven to significantly reduce the degeneration of catecholaminergic neurons in MPTP-treated monkeys and ameliorate LID in an MPTP-lesioned macaque model of PD [89,90]. Taken together, these data obtained from rodent and monkey models suggest that antagonists of mGluR5 are potential drugs that can reduce the degeneration of monoaminergic neurons in PD. The lower dosage combination of MPEP and the NMDA receptor antagonist MK-801 reversed motor deficits in 6-OHDA-lesioned rats, which suggests that the simultaneous blockade of mGluR5 and NMDA receptors may have more beneficial effects on PD treatment [91]. As a derivative of MPEP, Mavoglurant could increase the anti-Parkinsonian duration of levodopa in MPTP-lesioned monkeys [92]. Compared to MTEP and MPEP, Dipraglurant showed more promising effects in reducing overall LID severity and peak-dose dyskinesia in macaques [93,94]. In contrast to mGluR5, fewer studies focused on the assessment of mGluR1 as a potential target of anti-Parkinsonian actions. In a study comparing the efficiency of the mGluR5 antagonist MTEP and the mGluR1 antagonist (3-ethyl-2-methyl-quinoline-6-yl)-(4-methoxy-cyclohexyl)-methanone-methanesulfonate (EMQMCM) in different models of PD, the results showed that it was MTEP, but not EMQMCM, that inhibited levodopa-induced rotations and alleviated LIDs. This suggested that mGluR1 may not be an effective target for the symptomatic treatment of PD [95]. There is not a consensus about the exact mechanisms of the neuroprotective effects brought about by the antagonists of mGluR5. Except for the abovementioned interaction with NMDA receptors, the suppression of glial mGluR5 may decrease inflammatory damage and relieve MPTP-induced toxicity to midbrain dopaminergic neurons [96]. 

In contrast to mGluR5, it has been suggested that selective agonists or positive allosteric modulators of group II mGluR may contribute to PD treatment due to their presynaptic reduction of the corticostriatal transmission of glutamate, which is overstimulated in PD models. Consistent with this hypothesis, mGluR2/3 agonists (*S*)-(+)-alpha-amino-4-carboxy-2-methylbenzeneacetic acid (LY379268) attenuated akinesia in phencyclidine- and amphetamine-treated mouse models [97]. In addition, the inhibition of group II mGluRs using the antagonist LY341495 increased the evoked excitatory postsynaptic currents in rat SNc neurons [98]. Consistent with these observations, the group II mGluR agonists LY379268 and (2R,4R)-4-aminopyrrolidine-2,4-dicarboxylate (2R,4R-APDC) were proven to display neuroprotective effects through the reduction of the extent toxicity in a 6-OHDA-lesioned rodent PD model [99]. Similar results were observed in the combination treatment of LY379268 and the mGluR2/3 receptor agonist (2S,2’R,3’R)-2-(2’,3’-dicarboxycyclopropyl) glycine (DCG-IV) on 6-OHDA-lesioned rats with reduced corticostriatal transmission, providing support for group II mGluR agonists as potential neuroprotective drugs for PD treatment [100]. 

Similar to the findings for group II mGluR, accumulating evidence suggests that mGlu4 receptor activation may be beneficial for the treatment of PD. Systemic or intrapleural administration of the mGluR4 agonist Phenyl-7-(hydroxyamino) cyclopropopa[b] chrome-1a-carboxamide (PHCCC) contributed to the reduction of the extent of nigrostriatal toxicity induced by MPTP in wildtype mice, but not in mGluR4-deficient mice. This supports the potential of the selective activation of mGluR4 as a therapeutic strategy for the treatment of PD [101]. In addition, the mGluR4 positive allosteric modulator VU0364770 could potentiate the motor stimulation of a subthreshold L-DOPA dosage in 6-OHDA-lesioned rats [102]. Another mGluR4 positive allosteric modulator, VU0652957 (VU2957, valiglurax), has been processed to fabricate a spray-dried dispersion formulation for clinical application [103]. However, in contrast to mGluR4, the studies on the anti-Parkinsonian and neuroprotective properties of other group III mGluR subtypes are limited. One study of 6-OHDA-lesion rats showed that the intranigral infusion of the mGluR8-selective agonist (S)-3,4-dicarboxyphenylglycine (DCPG) induced mild catalepsy, which suggested that mGluR8 activation can reduce the anti-Parkinsonian effects induced by other group III mGluRs in the substantia nigra [104]. Therefore, subtype-selective agonists should be considered when taking group III mGluRs as targets for the treatment of PD.

## 4. Clinical Trials Targeting Glutamate Receptors in PD 

Though the obtained preclinical evaluations indicated the potential efficacy of targeting glutamate receptors for the treatment of parkinsonism, clinical trials in PD patients are still relatively insufficient and still under development. As of now, only the weak NMDA antagonist amantadine has become widely applied for the treatment of dyskinesias. The pharmacology of amantadine is complex and lacks a well-defined mechanism of action, yet it does exert an anti-Parkinsonian effect through “therapeutic” (i.e., low micromolar) concentrations on NMDA antagonist activity [105]. In a large retrospective series involving 836 Parkinsonian patients, amantadine treatment was shown to improve the survival of patients, which suggests its neuroprotective properties through NMDA receptor antagonism [106]. In addition, another uncompetitive NMDA receptor antagonist, dextromethorphan/quinidine, showed clinical benefits for the treatment of LID in a study involving 13 PD patients [107]. However, clinical treatment with other NMDA antagonists for PD patients is limited. One reason for this is that several NMDA antagonists with promising preclinical results failed in clinical trials owing to intolerable side effects, as complete glutamate antagonism may cause adverse cognitive effects. Therefore, comprehensive investigations of the characteristics of NMDA receptor subtypes and the use of more specific antagonists that do not result in a global receptor blockade could improve therapeutic efficacy while preventing the presence of side effects [108]. However, a recent study on the NR2B selective NMDA receptor antagonist MK-0657 suggested that a single dose of MK-0657 failed to improve LID and motor symptoms in PD patients [109]. In comparison to NMDA receptors, there are fewer clinical reports of targeting AMPA receptors for the treatment of PD. Perampanel, a selective AMPA receptor antagonist, has been used in two multicenter randomized, double-blind, placebo-controlled, parallel-group phase III studies. Although it showed better toleration and safety, no clinically significant improvement of levodopa-induced motor fluctuations was found [110,111,112]. In addition, talampanel has also been used in clinical studies to evaluate the efficacy and tolerability in levodopa-treated patients with PD, but no available data were obtained (NCT00108667). Although several drugs targeting AMPA receptors in recent clinical studies have assessed the efficacy of treating LID, the results of these studies are unavailable. More trials concerning the potential of AMPA receptor antagonists as adjunct therapies for improving levodopa treatment are required. In addition, a positive finding is that some mGluR5 antagonists, such as AFQ056-mavoglurant and ADX-48621-dipraglurant, have been assessed in human trials as anti-dyskinetic drugs. The obtained data prove the safety and good tolerance of these drugs, while no worsening PD motor symptoms occurred [94,113]. However, two randomized phase II studies showed that mavoglurant failed to improve LID in PD patients [114]. Therefore, further investigations with a larger number of patients are urgently need so that mGluR5 could be considered a potential target for PD treatment.

## 5. Conclusions

In the present review, we discussed the basic biological characteristics of glutamate receptors and their alterations in PD development and anti-Parkinsonian processes (summarized in Table 1). Glutamate receptors are distributed in almost all neural cell types. Subunit composition and expression variants significantly differ between different cell types and brain regions. Furthermore, the expression levels of subunits and variants change with the development of PD. Antagonists of NMDA receptors display beneficial effects on reversing motor symptoms, reducing LIDs, and slowing progressive neurodegeneration in preclinical PD models. While the results of targeting AMPA receptors are complex, their antagonists are effective in the treatment of LID, and their agonist displays the potential for neuroprotection. Pharmacological regulation of mGluRs shows even more potential for PD treatment because of their ability to fine-tune neurotransmission. The inhibition of mGluR5 and the activation of group II mGluRs, as well as mGluR4, have exhibited pharmacological efficiency in different animal models of PD. These therapy strategies (using the antagonists and agonist) can reverse motor disorder and provide a neuroprotective effect. However, in clinical practice, treatments targeting these glutamate receptors face some challenges due to the constant failure of previous treatments in PD [115]. Although the reasons for past failures require further discussion, some interesting possibilities have been investigated, such as ignoring the disease stage of patients [116], a lack of a basic disease mechanism, a lack of animal models with precise replicas of etiopathogenesis, uncertainty concerning drug dosing and binding site, and a lack of proper clinical trial methodology [117]. Moreover, some receptors act as a “double-edged sword”. Taking NMDA receptors as an example, their antagonists are usually used to target NMDA reporters. However, a recent study showed the reduction of the NMDA receptor co-agonist D-serine and the NMDA receptor subunits GluN1 and GluN2B in MPTP-lesioned macaques and the cerebrospinal fluid of PD patients [118]. Therefore, future investigations should consider the balance between receptor-related pro-death and pro-survival effects. In addition, traditional treatment mainly focuses on the improvement of motor dysfunction and ignores non-motor symptoms in PD, such as anxiety, apathy, cognitive dysfunction, and depression [119]. 

For all the above-outlined receptors, the selectivity of novel compounds is still the key procedure under development. This requires a further understanding of the precise roles of glutamate receptors in the regulation of basal ganglia circuitry both in normal physiological and pathophysiological conditions, as well as the development of the more selective compounds for certain subtypes so that rational designs for combination therapy can be formulated. Further promising studies on primate Parkinson’s disease models are necessary to determine the potential of clinical trials when combination therapy is employed to maximize efficacy and avoid the induction of unexpected side effects.

## Figures and Tables

**Figure 1 ijms-20-04391-f001:**
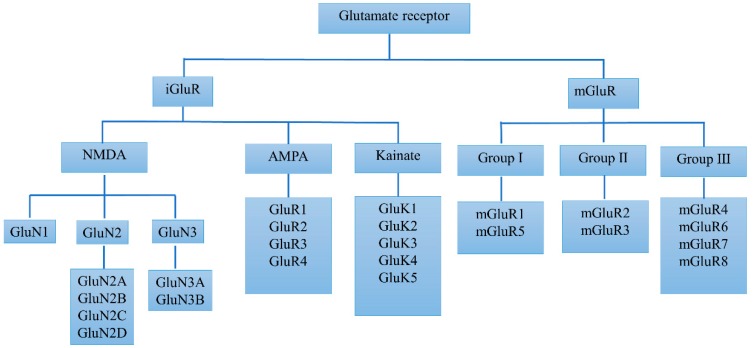
Classification of glutamate receptors.

**Table 1 ijms-20-04391-t001:** Alterations of glutamate receptors during PD and anti-Parkinsonian process.

Types	Changes during PD Process	Antiparkinsonian Treatment and Compounds	References
**NMDA**	GluN1, GluN2A, B, D: increase	NAMs: MK-801, dextrorphan, L-701324, SDZ220-581, MDL100,453, et al. GluN2B-selective NAMs: Ifenprodil, traxoprodil, Radiprodil Effects: improve PD motor symptoms, synergistically increase the anti-Parkinsonian efficiency of dopaminergic agents, reduce LIDs	[15,16,17,60,61,62,63,64,72,73,74,75]
**AMPA**	Increase	NAMs: NBQX, talampanel Effects: improve motor deficits, reduce LIDs	[23,77,78,79]
**KA**	GluK2: increase	NAMs: NBQX and CNQX (share with AMPA receptor), LU97175, LY382884, LY377770 Effects: no direct anti-Parkinsonian study targeting KAR	[32,83,84,85]
**mGluR1/5**	mGluR1: dynamically changed mGluR5: increase	NAMs of mGluR5: MPEP, MTEP, AFQ056, ADX48621 Effects: limite the extent of nigrostriatal damage, alleviate LIDs	[35,38,88,89,90,92,93,94]
**mGluR2/3**	mGluR2/3: decrease	PAMs: LY379268, (2R, 4R)-APDC, DCG-IV Effects: reduce the extent toxicity and corticostriatal transmission of 6-OHDA	[46,99,100]
**mGluR4,6,7,8**	lacking mGluR4 impairs learning ability of complex motor tasks	PAMs of mGluR4: PHCCC, ADX88178, VU0364770, Lu AF21934 Effects: reduce the extent nigrostriatal toxicity and dosage of L-DOPA	[101,102,120,121]

NAMs: Negative allosteric modulators; PAMs: Positive allosteric modulators.

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
