# Peer review of "Roles of Glutamate Receptors in Parkinson’s Disease"

_ijms, 2019, doi:10.3390/ijms20184391_

Round 1

Reviewer 1 Report

The manuscript presents an important topic and provides a comprehensive review. However, there are numerous problematic issues regarding general organization, style and grammar which need to be taken care of. The following are some examples:

Line 55 – change “glutamate play” to plays Line 69 – change “hydrolysis. while” to comma Line 81 – “we discuss regarding the distribution” Line 120 – change “Therefore, considerably evidence” to considerable Line 126 – “and localized in different” Line 169 – explain “loss of parkin function” Line 173 – change “mGluRs consist of mGluR1” to consists Line 174 – change “spines. while, in the primate” to While (capital). Also, consider removing “While” as it’s not necessary. Consider spacing paragraphs. For example: page 5 is almost entirely one paragraph. Line 401-402 – “restoring neurodegeneration”, confusing remark. Shouldn’t it be reversed?

I believe the document should be comprehensively edited before being accepted.

Author Response

Point 1: There are numerous problematic issues regarding general organization, style and grammar which need to be taken care of. The following are some examples: Line 55 – change “glutamate play” to plays Line 69 – change “hydrolysis. while” to comma Line 81 – “we discuss regarding the distribution” Line 120 – change “Therefore, considerably evidence” to considerable Line 126 – “and localized in different” Line 173 – change “mGluRs consist of mGluR1” to consists Line 174 – change “spines. while, in the primate” to While (capital). Also, consider removing “While” as it’s not necessary.

Response 1: Thanks for your suggestions. We send our manuscript to MDPI for English editing service checking grammar, spelling, punctuation and some improvement of style. And we also changed the words as suggested. Please refer to lines.

Point 2: Line 169 – explain “loss of parkin function”

Response 2: Parkin protein is encoded by PARK2 gene and widely distributed throughout the central nervous system. It was determined that mutations in parkin lead to autosomal recessive Parkinson's disease characterized by midbrain dopamine neuron loss [1]. This is suggested related to the regulation of excitatory synapses by Parkin since that knockdown of endogenous parkin triggers a proliferation of glutamatergic synapses and sensitizes in vitro midbrain dopaminergic neurons to kainate toxicity [2, 3]. Further mechanistic study showed that parkin can ubiquitinate kainate receptor (KAR) GluK2 and loss of parkin increases KAR currents and excitotoxicity [4]. This suggest that KAR upregulation may have a pathogenetic role in parkin-related autosomal recessive juvenile parkinsonism.

Point 3: Line 173 – change “mGluRs consist of mGluR1” to consists Line 174 – change “spines. while, in the primate” to While (capital). Also, consider removing “While” as it’s not necessary.

Response 3: Thanks for your suggestion. We removed it.

Point 4: Consider spacing paragraphs. For example: page 5 is almost entirely one paragraph

Response 4: Thanks for your suggestion. We have spaced it into 3 paragraphs according to three group of mGluRs

.

Point 5: Line 401-402 – “restoring neurodegeneration”, confusing remark. Shouldn’t it be reversed?

Response 5: Thanks for your suggestion. After checking, we have changed “restoring neurodegeneration” into “slowing progressive neurodegeneration”.

We also tried our best to improve the manuscript and made some changes in the manuscript, which will not influence the content of the paper.

 Reference:

[1]. Kitada, T.; Asakawa, S.; Hattori, N.; Matsumine, H.; Yamamura, Y.; Minoshima, S.; at al. Mutations in the parkin gene cause autosomal recessive juvenile parkinsonism. Nature 1998, 392, 605–608.

[2]. Helton, T.D.; Otsuka, T.; Lee, M.C.; Mu, Y.; Ehlers, M.D. Pruning and loss of excitatory synapses by the parkin ubiquitin ligase. Proc Natl Acad Sci 2008, 105, 19492–19497.

[3]. Staropoli, J.F.; McDermott, C.; Martinat, C.; Schulman, B.; Demireva, E.; Abeliovich, A. Parkin is a component of an SCF-like ubiquitin ligase complex and protects postmitotic neurons from kainate excitotoxicity. Neuron 2003, 37, 735–749.

[4]. Maraschi, A.; Ciammola, A.; Folci, A.; Sassone, F.; Ronzitti, G.; Cappelletti, G.; et al. Parkin regulates kainate receptors by interacting with the GluK2 subunit. Nat Commun 2014, 5, 5182.

Reviewer 2 Report

The propose review is dealing with Glutamate receptors involvement and an overview of the therapeutical approaches to modulate their functions in Parkinson’s disease. 

The overall review is interesting and constitutes a convincing overview of the existing litterature. However, the  proposed manuscript needs a thorough revision and an extensive editing of English language and style. Furthermore, a figure summarizing the overall review would be appreciated in addition to the proposed classification of glutamate receptors.

Author Response

Point 1: However, the proposed manuscript needs a thorough revision and an extensive editing of English language and style.

Response 1: Thanks for your suggestions. We send our manuscript to MDPI for English editing service checking grammar, spelling, punctuation and some improvement of style.

Point 2: Furthermore, a figure summarizing the overall review would be appreciated in addition to the proposed classification of glutamate receptors.

Response 2: Thanks for your suggestions. Because glutamate receptors contain various subunits and involve different drugs for treatment, it is difficult to summary them clearly in one figure. Alternatively, we summarized them in a table (Table 1 in revised   manuscript), which is more concise.

We also tried our best to improve the manuscript and made some changes in the manuscript, which will not influence the content of the paper.

Round 2

Reviewer 2 Report

I would thank the authors to have followed all the recommendations.

The revised version is acceptable for publication.